# Tissue Sampling and Homogenization with NIRL Enables Spatially Resolved Cell Layer Specific Proteomic Analysis of the Murine Intestine

**DOI:** 10.3390/ijms23116132

**Published:** 2022-05-30

**Authors:** Hannah Voß, Manuela Moritz, Penelope Pelczar, Nicola Gagliani, Samuel Huber, Vivien Nippert, Hartmut Schlüter, Jan Hahn

**Affiliations:** 1Section/Core Facility Mass Spectrometry and Proteomics, University Medical Center Hamburg-Eppendorf (UKE), Martinistraße 52, 20246 Hamburg, Germany; ha.voss@uke.de (H.V.); ma.moritz@uke.de (M.M.); vivien.nippert@gmail.com (V.N.); 2Section of Molecular Immunology und Gastroenterology, I. Department of Medicine, University Medical Center Hamburg-Eppendorf (UKE), Martinistraße 52, 20246 Hamburg, Germany; p.pelczar@uke.de (P.P.); n.gagliani@uke.de (N.G.); s.huber@uke.de (S.H.); 3Department of General, Visceral and Thoracic Surgery, University Medical Center Hamburg-Eppendorf (UKE), Martinistraße 52, 20246 Hamburg, Germany

**Keywords:** tissue sampling, nanosecond infrared laser, laser ablation, proteomics, mass spectrometry, colon tissue, 3D-sampling, miniaturization

## Abstract

For investigating the molecular physiology and pathophysiology in organs, the most exact data should be obtained; if not, organ-specific cell lines are analyzed, or the whole organ is homogenized, followed by the analysis of its biomolecules. However, if the morphological organization of the organ can be addressed, then, in the best case, the composition of molecules in single cells of the target organ can be analyzed. Laser capture microdissection (LCM) is a technique which enables the selection of specific cells of a tissue for further analysis of their molecules. However, LCM is a time-consuming two-dimensional technique, and optimal results are only obtained if the tissue is fixed, e.g., by formalin. Especially for proteome analysis, formalin fixation reduced the number of identifiable proteins, and this is an additional drawback. Recently, it was demonstrated that sampling of fresh-frozen (non-fixed) tissue with an infrared-laser is giving higher yields with respect to the absolute protein amount and number of identifiable proteins than conventional mechanical homogenization of tissues. In this study, the applicability of the infrared laser tissue sampling for the proteome analysis of different cell layers of murine intestine was investigated, using LC–MS/MS-based differential quantitative bottom-up proteomics. By laser ablation, eight consecutive layers of colon tissue were obtained and analyzed. However, a clear distinguishability of protein profiles between ascending, descending, and transversal colon was made, and we identified the different intestinal-cell-layer proteins, which are cell-specific, as confirmed by data from the Human Protein Atlas. Thus, for the first time, sampling directly from intact fresh-frozen tissue with three-dimensional resolution is giving access to the different proteomes of different cell layers of colon tissue.

## 1. Introduction

The analysis of the entirety of biomolecule classes—omes—in cultured cells in the past two decades has largely improved our knowledge about molecular physiology and pathophysiology. However, the limits of cell-culture-based research is getting more and more obvious. For circumventing these limits, the development of organoids is currently a fast-growing field in life sciences [1]. Nevertheless, the ideal case is to study omes in the cells of tissues because, even in organoids, the microenvironment is lacking [1].

Until today, the analysis of omes from defined cells of a tissue of interest is still challenging and comprises several steps. After removing the tissue from the organism, it must be homogenized for releasing and solubilizing molecules from the tissue in a liquid for making them accessible to analytical techniques. After solubilization, the biomolecules of interest may pass sample preparation steps, e.g., for their enrichment, prior to their analysis with methods such as liquid chromatography (LC) and mass spectrometry (MS). The homogenization of the tissue is usually achieved mechanically. Several homogenization procedures are available and chosen with respect to the type of tissue, as well as the subsequent type of bioanalytical method and biomolecule class of interest (e.g., proteins [2], nucleotides [3], and lipids [4]. Common procedures are, for example, dispersing machines using the rotor-stator principle, milling devices for transforming freeze-dried tissue to powders, bead mills for small amounts of fresh tissue (“wet milling”) [5], or cryogenic grinding devices [6]. High hydrostatic pressure was demonstrated for tissue homogenization by Gross et al. [7].

Homogenization of tissue is always a critical step for several reasons: (1) Homogenization may not be complete, thus leaving biomolecules in those parts, which are not solubilized, and which are part of the pellets remaining after centrifugation. In addition, the insoluble materials may adsorb other biomolecules and, thus, be co-precipitated. Biomolecules being integrated into these insoluble parts or adsorbed to them are lost for subsequent analysis. (2) During homogenization, cells are broken, and, thereby, enzymes are released from their compartments, such as lysosomes. Enzymes can quickly degrade or convert other biomolecules. Proteases are cleaving other proteins; phosphatases are removing phosphates from phosphor-proteins and other phosphorylated molecules. For all biomolecule classes from a cell, there are enzymes that can convert them. 

(3) Biomolecules released from their compartments are not protected anymore and therefore often prone to chemical reactions such as oxidation [8], reactions with other biomolecules [9], or with buffer components [10]. Such chemical reactions can already occur during homogenization of tissue. As a result of all these problems that occur during homogenization, the original composition of biomolecules is changed, yielding false results, which make the deciphering of cell physiological mechanisms even more difficult or impossible.

Diverse procedures and instruments were developed for minimizing the above-mentioned problems. For example, protease inhibitors are used for decreasing the enzymatic degradation of proteins [2]. Detailed insights into problems associated with homogenization of tissues and actions for decreasing these problems are given in numerous reviews, e.g., from Bodzon-Kulakowska et al. [11], Gli et al. [12] and Goldberg et al. [6]. An effective instrument to significantly decrease enzymatic affinities in tissues was developed by Svensson et al. [13]. This instrument rapidly and irreversibly denatures the proteins in a tissue sample by conductive heating under controlled pressure This commercially available instrument (“denator”) is a step forward regarding the stabilization of biomolecules and has been shown in many investigations [14] and summarized in several reviews (e.g., Sköld et al.) [15]. However, the denator instrument is not a solution regarding the problems associated with homogenization as described above.

The state-of-the-art is still the mechanical homogenization of tissues for subsequent omics, dominated by cryo-pulverization procedures. Critical in almost all mechanical homogenization techniques is the problem that the tissue is not 100% solubilized. After centrifugation, there will be still a pellet with undissolved tissue material. A disadvantage is that, in the investigations mentioned above, tissues are homogenized in toto without considering their composition comprising different cell types. 

An important aspect of tissues is their three-dimensional spatial organization of tissue-specific cell types. The high importance of spatially and cell-type resolved omics analysis of tissue has been shown, for example, by Doll et al. [16], who described significant proteomic differences across different cell types and regions within the heart, brain, and liver. This aspect should also be carefully addressed, especially in the investigation of tumor tissue.

It is well-known, as Mertins et al., 2018 [17] stated, that solid tumor tissues typically consist of at least epithelial, stromal, and hematologic components, as well as necrotic areas. Thus, for characterizing the omes of cancer cells in a solid tumor, it is desirable to differentiate between the different cancer-associated cell types and exclude non-cancer cell types that are present near the tumor. In the worst case, the important markers of cancer cannot be recognized anymore. A further problem, decreasing the number of identifiable proteins, originates from proteins derived from blood in tissues. Blood-borne proteins have strong suppressive effects in mass spectrometry. Thus, a technique offering spatial resolution during sampling is very advantageous if it is the aim to characterize the entirety of a biomolecule class. 

The spatial aspect of tissues today is best addressed by laser-capture microdissection (LCM) [18]. Cells of interest in a tissue section are recognized by a microscope, cut out by a laser, and then collected for subsequent analysis. Hunt et al. (2021) [19] demonstrated the benefits of investigating tumor tissue applying LCM in a 3-dimensional fashion. In this manner, detailed insights into tumor microenvironment heterogeneity are obtained, thereby getting access to prognostic signatures (biomarkers) and molecular cancer pathophysiology [19]. However, LCM is very time-consuming. The best results regarding the selection of target cells are obtained if the tissue is fixed by formalin, where the morphologic structure of the tissue is best preserved, guaranteeing high-quality microscopic images. For proteomics, formalin fixation results in a limited extraction efficiency, as well as the induction of irreversible chemical modification, finally leading to higher error rates and reduced protein numbers in protein identification from LC–MS/MS measurements [20].

A technique, overcoming several of the problems mentioned above, is the sampling and homogenization of tissue with an infrared (IR) laser. Scanning tissue with picosecond IR laser (PIRL) or nanosecond IR laser (NIRL) rapidly converts the tissue into an aerosol by soft cold vaporization. The tissue aerosol is already a homogenate. Additional homogenization is not required. A liquid condensate of the tissue aerosol after centrifugation shows no pellet. 

Kwiatkowski et al. showed that, compared to mechanical homogenization, the yield and number of identified proteins increased significantly when a picosecond infrared laser (PIRL) was used [21]. Besides PIRL [21,22,23] and microsecond infrared laser (MIRL) [23]. especially nanosecond infrared lasers (NIRL) with a pulse duration of about 7 ns, have been successfully utilized for tissue sampling prior to proteome analysis [24,25,26]. In the recent study by our group, Hahn et al. [26] revealed the clear distinguishability of murine colon and spleen tissue, based on >1000 identified proteins after collecting NIRL-induced tissue aerosols as condensates on glass cover slips, followed by bottom-up proteomics. 

In this study, we used NIRL-based sampling to investigate the specific proteomes of different tissue cell layers of the intestine, using LC–MS/MS-based differential quantitative proteomics and following the hypothesis that cell-type-specific proteins are identified in the different colon cell layers. 

## 2. Results

Two laser sampling studies of murine intestine tissue were performed to determine the applicability of NIRL ablation for three-dimensional sampling prior to quantitative differential proteomics.

### 2.1. Analysis of the Proteomes of Different Colon Regions

With the subsequent proteomic analysis of samples obtained by NIRL ablation of ascending (A), transversal (B), and descending (C) colon (Figure 1a), 3053 proteins were quantified (*n* = 3), yielded from an ablation area of about 5 mm^2^; and an ablation depth of 117 µm (approximately 0.6 µL).

In total, 303 proteins were found to be, via ANOVA, significantly different (*p*-value < 0.05) (Figure 1b and Appendix A). KEGG-pathway-based enrichment analysis of these proteins revealed that they were predominantly associated with metabolic processes (Figure 1c). An enrichment analysis based on gene ontology further indicated a predominant location of regulated proteins in extracellular exosomes (Figure 1d), as well as an enrichment of proteins with antioxidant activity among other terms (Figure 1c–e and Appendix A).

### 2.2. Analysis of the Proteomes of Consecutive Colon Tissue Layers

Eight consecutive layers were sampled with NIRL and analyzed with differential quantitative proteomics, as shown in Figure 2a.

In this study, a total of 2882 proteins were quantified (Appendix A). Nonlinear-iterative-squares PCA revealed a separation of all ablated layers based on 2474 proteins that were found in at least 50% of all samples. Ablation layers formed a consecutive gradient along principal component (PC) one, accounting for 35% of the explained variance (Figure 2b and Appendix A). PC2, accounting for 30% of the explained variance, represented differences between the adjacent areas of the colon. To evaluate how many distinguishable cellular areas were represented across the ablated layers, Pearson correlation-based consensus clustering was applied (Appendix A). To determine the optimal number of clusters, the proportion of ambiguous clustering (PAC) score was calculated, and we identified four differentiable protein patterns across the ablated layers (Appendix A). The first area included the ablated layers one and two (0–234 µm). The second area comprised the ablated layers three and four (234–468 µm). Ablated layer five formed an individual area (468–585 µm). Based on the protein abundance distribution, a common area was defined for ablated layers 6–8 (585–939 µm). A total of 251 proteins were identified as statistically differential abundant across different cellular areas in ANOVA testing (*p*-value < 0.05) (Appendix A). While clearly distinguishable proteome profiles were formed between cellular areas one and four. The Pearson-correlation-based hierarchical clustering revealed that the protein profiles of areas two and three showed high abundances of cell-type-specific proteins found in both areas one and four, respectively (Appendix A). The Student’s *t*-test revealed 164 statistically significant higher-abundance proteins for ablated area one and 103 higher abundant proteins for area four, respectively. For area two, seven higher-abundance proteins were identified, while only three showed an elevated intensity in area three (Appendix A).

To further determine the predominate cellular identity of areas one and four, gene ontology biological processes (GO-BP)-based gene-set enrichment was performed, comparing the proteome profile of areas one and four to all other cellular areas (Figure 2e,f and Appendix A). The gene-set enrichment analysis (GSEA) revealed an enrichment of proteins associated with metabolic processes, digestion, lipid metabolism, peptidase activity, and cation homeostasis (Figure 2c). Furthermore, leucocyte (Figure 2e) and differentiation of epithelial-cell-specific proteins showed a higher abundance (Figure 2f) in cellular area one, decreasing toward cellular area four.

For cellular area four, an enrichment of proteins associated with muscle development, muscle contraction, morphogenesis, mRNA processing, chromosomal organization, and epithelial to mesenchymal transition was found (Figure 2d and Appendix A). Based on the results of GSEA, a representation of the mucosa colon layer, including the leucocyte-infiltrated epithelium, can be suspected for the first two ablation layers. Furthermore, it can be assumed that ablation layers six to eight reflect the muscularis propria.

To further test this hypothesis, the abundance of established epithelial (*SFN, KRT18, VIL1, KRT20,* and *CDH1*) (Figure 3a) and muscular marker proteins (*MYL9, SMTN, CNN1, ACTA1,* and *TPM2*) (Appendix A) across different ablation layers was analyzed and compared to immunohistological (IHC) staining of the colon obtained from the Human Protein Atlas (v21.0.proteinatlas.org) [27].

For all epithelial marker proteins, the highest abundance was identified in the first ablation layer and gradually decreased toward the last ablation layer. These findings go in line with the IHC staining intensity of respective proteins in the human colon epithelium obtained from the Human Protein Atlas (v21.0.proteinatlas.org) [27]. For the ablation layers 6–8, a higher abundance of muscular marker proteins was found. IHC staining additionally confirmed a higher abundance of these proteins in the muscularis propria layer of the colon obtained from the Human Protein Atlas (v21.0.proteinatlas.org) [27].

## 3. Discussion

In this study, we applied consecutive NIRL-based sampling of colon for obtaining 3D-resolved samples for characterizing the proteomes of different specific areas of the tissue. In this way, we demonstrate the applicability of NIRL-based spatial sampling of tissues at lateral and in-depth resolution.

Addressing the importance of the localization of cells in organs, significant differences were shown between ascending (AC), descending (DC), and transversal (TV) colon. These differences between protein profiles of the epithelial cells of the mucosa of different areas of the colon are mostly assigned to metabolic processes (glycolysis, TCA cycle, amino acid, and sugar metabolism) and antioxidant activities. The predominant functions of the gastrointestinal tract are digestion, absorption of nutrients, secretion, and immune response and control [28]. Differences in the metabolome of AC and DC colon tissue, including the profiles of nucleotides, amino acids, and lipids, have been described for healthy, overweight, and obese adults, respectively [29]. To our knowledge, no study comparing the proteomic profiles of colon tissue in AC, DC, and TV colon has currently been performed. However, the previously described findings at the metabolome level are in line with the changes at the proteome level described here and underline the successful laterally resolved proteome analysis of colon tissue after NIRL-based sampling. 

In comparison to our most recent study [26], where 1617 proteins were identified from murine colon tissue, using a similar setup, we demonstrated a significant increase in the number of identified proteins (3053) for a comparable ablation volume (approximately 0.6 µL). This increased efficiency can be explained by direct trapping of the aerosol on a glass slide above the tissue sample and reducing the scanning area for the laser ablation by shifting of sample repeatedly. Since the scanning laser beam has to pass the glass slide, this scanning area is always lost (Appendix A). Furthermore, we increased the precision of the laser ablation by introducing a scan lens and synchronized laser triggering to the setup. 

In the second part of the study, we aimed to investigate the applicability of NIRL-based tissue sampling for a three-dimensional (3D) analysis of colon tissue. Here, eight consecutive layers were ablated, covering a total depth of 936 µm in approximately 117 µm steps, to disclose the proteome of different cell layers of the intestine. Thereby, we obtained four different cell-layer-specific areas according to the proteome profiles of the eight different samples yielded by consecutive laser ablation. Hence, ablation layers one and two showed high abundances of epithelial- and leucocyte-specific proteins, indicating a representation of the mucosa cell layer of the intestine at an ablation depth of 0–234 µm, which predominantly consists of epithelial and immunological cells [28,30]. The higher abundance of muscle-cell-specific proteins in ablation layers 6–8 revealed a depiction of the muscularis propria at 585–936 µm, consisting of a large layer of smooth muscle cells arranged in parallel arrays [28]. Between 234 and 585 µm, two different cellular areas of the muscularis mucosa and submucosa were identified that showed high intensities of both muscle- and epithelium-specific proteins, respectively. This can be explained by the cellular architecture of the colon showing epithelial enclosures across all cellular layers, except for the muscularis propria [31]. Muscle-cell-specific profiles at respective ablation depths are represented due to the presence of smooth muscle cells in the muscularis mucosa [28].

While different cellular areas of the intestine could be differentiated, a single-cell-type resolution was not yet achieved in this study, which would particularly be important for studying the microenvironment of murine intestine in more detail. Nevertheless, NIRL ablation enabled a spatially resolved analysis of the colon, which was not possible before our study, with an in-depth resolution of about 117 µm.

In summary, we have shown for the first time that infrared laser ablation of tissue enables spatially resolved analysis of cell-layer-specific proteomes directly from intact fresh-frozen organs. While this technique successfully addresses the spatial aspect of tissues, in contrast to other approaches, NIRL ablation does not require additional steps to homogenize the tissue or extract the proteins, making it suitable for proteome analysis on a miniaturized scale.

## 4. Material and Methods

### 4.1. Animals

Mice aged 8–12 weeks old used in this study were on a C57/BL6 background. Mice were kept under specific pathogen free conditions, at an ambient temperature of 20 ± 2 °C, humidity of 55 ± 10%, and a dark/light cycle of 12 h.

### 4.2. Ablation Setup

The experimental ablation setup is shown in Figure 4a. 

The beam of a nanosecond infrared laser system (LS) (Opolette SE 2731, Opotek, LLC, Carlsbad, CA, USA) with a pulse width of 7 ns and a tunable wavelength (2.70–3.10 µm) is widened and collimated by a Galaien telescope, consisting of a concave and a convex lens (ISP-PC-25-75 and ISP-PX-25-150, ISP Optics Latvia, LTD, Riga, Latvia). An f-theta scanning lens (SL1-2.94-36-10-100-U-A, II–VI, Inc., Saxonburg, PA, USA) with 100 mm focal length in combination with a 2-axis scanning mirror (OIM202, Optics in Motion LLC, Long Beach, CA, USA) is used to scan the frozen sample, located on a manual xy-stage (XR25P-K1/M, Thorlabs, Newton, NJ, USA) with a custom-built cooling system. The computer synchronizes the scanning mirror with the laser system, using two analog lines and a digital trigger line of an input/output card (USB-6343, National Instruments, Austin, TX, USA). A glass slide (SuperFrost^®^ microscope slides, R. Langenbrinck GmbH, Emmendingen, Germany) on a manual xyz-stage (XR25P-K2/M, Thorlabs, Newton, NJ, USA) set is mounted closely above the ablation site to collect the aerosol. The ablation can be observed with a microscope camera (DFK 23UP031, The Imaging Source, Bremen, Germany) through a dichroitic mirror (Separator 109093, Layertec GmbH, Jena, Germany). 

### 4.3. Tissue Sampling and Collection Procedure

For the study on murine colon, the laser wavelength was set to 2.85 µm. A preliminary study showed a maximum protein amount, determined with BCA tests, at this wavelength for this tissue type. The pulse energy was measured to be 0.9 mJ at sample position, corresponding to a fluence of 18.1 J/cm^2^ with focal spot dimensions of about dx = 125 µm and dy = 100 µm in diameter, respectively.

Positively charged glass slides (SuperFrost^®^, microscope slides, R. Langenbrinck GmbH, Emmendingen, Germany) are placed directly above the tissue at the ablation area to trap the resulting aerosol with the charged side facing the sample (Figure 4c,d). To increase yield of the trapped aerosol, the beam scanning area was reduced to 1 × 0.5 mm^2^; (Figure 4d, black arrow). Optimal laser focus positioning was achieved by synchronizing a two-axis scanning mirror in combination with a f-theta scan lens of 100 mm focal length with the laser pulse triggering, utilizing a fast input/output card. The custom-made control software enables high-precision placement of the focal laser spot by introducing wait times to let the scanning mirror reach its position and swing out. The integrated monitoring camera with zoom objective was used for calibrating and monitoring lateral spot positioning. Optimal axial positioning of the sample was achieved by observing the dimensions of laser shots on photo paper at different z-positions with reduced laser energy.

After each ablation of a layer set, the sample stage was manually shifted laterally (x-direction) for 600 µm to extend the ablation area to about 5 mm^2^.

After completion of the sampling, the dried aerosol was dissolved with 100 µL of buffer (0.1 M triethylammonium bicarbonate buffer with 1% sodium deoxycholate), using four pipetting steps with 25 µL to reduce the contact area of the protein solution with the glass surface. The removed glass slide was then replaced with a new one for the next layer.

Using this optimized laser setup, two laser ablation experiments were performed in the murine intestine, while obtaining the spatial information of the sample: In the first experiment, one layer was ablated at ascending (A), transversal (B), and descending (C) colon, respectively.In the second experiment, 8 consecutive layers were ablated.

### 4.4. Determination of Layer Thickness

Layer thickness was determined by utilizing a spectral domain optical coherence tomography system (TEK221PSC2-SP1, Thorlabs, Newton, NJ, USA) with a central wavelength of 1300 nm and the imaging volume set to 699 × 699 × 768 voxels, each measuring 8.58 × 8.58 × 3.45 µm³ in air. The measurements were performed on three ablations on an additional frozen colon at three different positions (Ab1, Ab2, and Ab3) (Figure 5a–c). Ablation depth was determined (see Figure 5c) at each of the ablation sites, based on 10 random distance measurements, revealing depths of 226 ± 17 µm, 245 ± 14 µm, and 203 ± 20 µm, respectively. The mean layer depth over all ablation sites was determined to about 117 µm. This value was used for all further considerations.

The ablation parameters for spacing were kept to 143 µm × 125 µm, the ablation area was reduced to 1 mm × 1 mm, and the number of ablation layers was reduced to 2. First the OCT image data sets (before and after ablation) were preprocessed (window/level, blur filter, cropped) with Fiji (ImageJ 1.53f51) [32] and afterward registered with the Fiji plugin Fijiyama (version “handsome honeysuckle”) [33] to generate an overlay image stack and determine ablation volume dimensions. For volume quantification and mean depth determination (Appendix A), the open-source software ITK-Snap (Version 3.8.0 [34]) with its onboard measuring tools was used.

### 4.5. Tryptic Digestion of Proteins of the Ablated Tissue Samples

The dissolved aerosols were heated and mixed 10 min at 99 °C, at a rotation speed of 400 rpm, using a shaker (ThermoMixC^®^, Eppendorf SE, Hamburg, Germany). Sonification was performed for 5 pulses at 30% power. Disulfide bonds were reduced with 10 mM DTT at 60 °C for 30 min. Cysteine residues were alkylated with 20 mM iodoacetamide (IAA) for 30 min at 37 °C in the dark. Tryptic digestion was performed for 16 h at 37 °C, using 250 ng trypsin. After tryptic digestion, the inhibition of trypsin activity, as well as the precipitation of sodium deoxycholate, was achieved by the addition of 1% formic acid (FA). Samples were centrifuged for 5 min at 14,000× *g*. The supernatant was collected, dried in a vacuum concentrator centrifuge (UNIVAPO 100 H, UniEquip, Martinsried, Germany), and stored at −20 °C until further use. Prior to mass spectrometric analyses, peptides were resuspended in 10 µL of 0.1% FA.

### 4.6. LC–MS/MS Data Acquisition

Liquid chromatography–tandem mass spectrometer (LC–MS/MS) measurements were performed on a quadrupole-ion-trap-orbitrap MS (Orbitrap Fusion, Thermo Fisher Scientific, Waltham, MA, USA) coupled to a nano-UPLC (Dionex Ultimate 3000 UPLC system, Thermo Fisher Scientific, Waltham, MA, USA). Tryptic peptides were injected to the LC system via an autosampler, purified and desalted by using a reversed phase trapping column (Acclaim PepMap 100 C18 trap; 100 μm × 2 cm, 100 A pore size, 5 μm particle size; Thermo Fisher Scientific, Waltham, MA, USA), and thereafter separated with a reversed phase column (Acclaim PepMap 100 C18; 75 μm × 50 cm, 100 A pore size, 2 μm particle size, Thermo Fisher Scientific, Waltham, MA, USA). Trapping was performed for 5 min at a flow rate of 5 µL/min with 99% solvent A (0.1% FA) and 1% solvent B (0.1% FA in ACN). Separation and elution of peptides were achieved by a linear gradient from 1 to 30% solvent B in 70 min at a flow rate of 0.3 µL/min. Eluting peptides were ionized by using a nano-electrospray ionization source (nano-ESI) with a spray voltage of 1800 V, transferred into the MS and analyzed in data dependent acquisition (DDA) mode. For each MS1 scan, ions were accumulated for a maximum of 120 ms or until a charge density of 2 × 10^5^ ions (AGC target) was reached. Fourier-transformation-based mass analysis of the data from the orbitrap mass analyzer was performed by covering a mass range of 400–1300 *m*/*z* with a resolution of 120,000 at *m*/*z* = 200. Peptides with charge states between 2+–5+ above an intensity threshold of 1000 were isolated within a 1.6 *m*/*z* isolation window in top-speed mode for 3 s from each precursor scan and fragmented with a normalized collision energy of 30%, using higher energy collisional dissociation (HCD). MS2 scanning was performed, using an ion trap mass analyzer, covering a mass range of 380–1500 *m*/*z* with an orbitrap resolution of 15,000 at *m*/*z* = 200 and accumulated for 60 ms or to an AGC target of 1 × 10^5^. Already fragmented peptides were excluded for 30 s

### 4.7. Raw Data Processing

Raw data from LC–MS/MS measurements were processed with MaxQuant (version 1.6.2.10, Max Planck Institute for Biochemistry, Martinsried, Germany), using the integrated Andromeda algorithm. For protein identification, measured MS2 spectra were searched against theoretical fragment-spectra of tryptic peptides, generated from a reviewed murine Swiss-Prot FASTA database obtained in February 2020 containing 17,015 entries. All samples were handled as individual experiments. The carbamethylation of cysteine residues was set as a fixed modification. methionine oxidation, protein N-terminal acetylation, removal of the initiator methionine at the protein N-terminus and the conversion of glutamine to pyroglutamate were set as variable modifications. Peptides with a minimum length of 6 amino acids and a maximum mass of 6000 Da were identified with a mass tolerance of 10 ppm. Only peptides with a maximum of two missed trypsin cleavage sites were considered. For peptide identification, matching between runs was applied, using a match time window of 0.7 min and an alignment time window of 20 min between individual runs. The error tolerance was set to 20 ppm for the first precursor search and to 4.5 ppm for the following main search. Fragment spectra were matched with 20 ppm error tolerance. A false discovery rate (FDR) value threshold <0.01, using a reverted decoy peptide database approach was set for peptide identification. Label free quantification was performed with an LFQ minimum ratio count of 1. For quantification, all identified razor and unique peptides were considered. The label minimum ratio count was set to 1. 

### 4.8. Data Analysis and Visualization

Identified protein group abundances were loaded into Perseus (version 1.6.15.0, Max Planck Institute for Biochemistry, Martinsried, Germany), log2 transformed and normalized to the total protein amount per sample to compensate for variations in the injected sample amount. For the analysis of the proteomes of consecutive colon layers, one replicate of layer 3 (depth of 234–351 µm) was excluded due to high yields of blood proteins. Hence, for further statistical testing layer three was excluded due to the absence of replicates.

For ANOVA and t-testing a *p*-value cutoff < 0.05 was set. *p*-value significant proteins with a foldchange difference <1.5 were considered for further analysis after t-testing. 

Nonlinear Iterative vertical Least Squares (NIPALS) PCA and hierarchical clustering were performed in the R software environment (version 4.1.3). For Principal component calculation and visualization, the mixOmics package was used in Bioconductor (version 3.14) [35] Hierarchical clustering was performed based on centered and normalized abundances using the heatmap package. Pearson correlation was applied as a distance metric. Ward.D linkage was applied. Missing data in correlation distances was pairwise complete correlation.

To determine the number of distinguishable molecular clusters between different ablation layers, consensus clustering was applied, using the ConsensusClusterPlus package [36]. Pearson correlation was applied as a distance metric. Ward.D linkage was used. Pairwise complete correlation was used to deal with missing values in the generation of Pearson correlation matrices. The number of subsamples was set to 1000 M maximum number of 7 clusters, corresponding to the total number of ablation layers that were used. Reactome-based [37] Gene Set Enrichment Analysis (GSEA) was performed by using the software GSEA (version 4.1, Broad Institute, San Diego, CA, USA), [38] based on log2-transformed normalized protein abundances. One hundred permutations were used. Permutation was performed based on gene sets. A weighted enrichment statistic was applied, using the signal-to-noise ratio as a metric for ranking genes. No additional normalization was applied within GSEA. Gene sets smaller than 15 and bigger than 500 genes were excluded from analysis. For visualization of GSEA results, the Cytoscape environment (version 3.8.2) [39] was used. The EnrichmentMap (version 3.3) [40] application was used. Gene sets were considered if they were identified at an FDR < 0.25 and a *p*-value < 0.1. For gene-set-similarity filtering, data set edges were set automatically. A combined Jaccard and Overlap metric was used, applying a cutoff of 0.375. For gene set clustering, AutoAnnotate (version 1.3) [41] was applied. A cluster algorithm Markov cluster algorithm (MCL) was used. The gene-set-similarity coefficient was utilized for edge weighting. Individually analyzes gene sets (leucocyte mediated immunity; keratinocyte/epithelial signatures) were obtained from the molecular signature database (version 3.0) [42].

For the enrichment analysis of ANOVA significant proteins, the DAVID functional annotation tool (version 6.8) [43] was used based on all gene ontology (GO) terms [44]. Terms enriched with a *p*-value < 0.01 were considered significantly enriched. 

### 4.9. Histological Staining

Immunohistological stainings of the human intestine for selected muscular and epithelial marker proteins were obtained from the Human Protein Atlas (v21.0.proteinatlas.org) [27]. The following images were used: *SFN* (https://www.proteinatlas.org/ENSG00000175793-SFN/tissue/colon#img), accessed on 29 April 2022;*KRT18* (https://www.proteinatlas.org/ENSG00000111057-KRT18/tissue/colon#img), accessed on 29 April 2022;*VIL1* (https://www.proteinatlas.org/ENSG00000127831-VIL1/tissue/colon#img), accessed on 29 April 2022;*KRT20* (https://www.proteinatlas.org/ENSG00000171431-KRT20/tissue/colon#img), accessed on 29 April 2022;*CDH1* (https://www.proteinatlas.org/ENSG00000039068-CDH1/tissue/colon#img), accessed on 29 April 2022;*TPM2* (https://www.proteinatlas.org/ENSG00000198467-TPM2/tissue/colon#img), accessed on 29 April 2022;*ACTA1* (https://www.proteinatlas.org/ENSG00000143632-ACTA1/tissue/colon#img), accessed on 29 April 2022;*CNN1* (https://www.proteinatlas.org/ENSG00000130176-CNN1/tissue/colon#img), accessed on 29 April 2022;*SMTN* (https://www.proteinatlas.org/ENSG00000183963-SMTN/tissue/colon#img), accessed on 29 April 2022;*MYL9* (https://www.proteinatlas.org/ENSG00000101335-MYL9/tissue/colon#img), accessed on 29 April 2022.

## Figures and Tables

**Figure 1 ijms-23-06132-f001:**
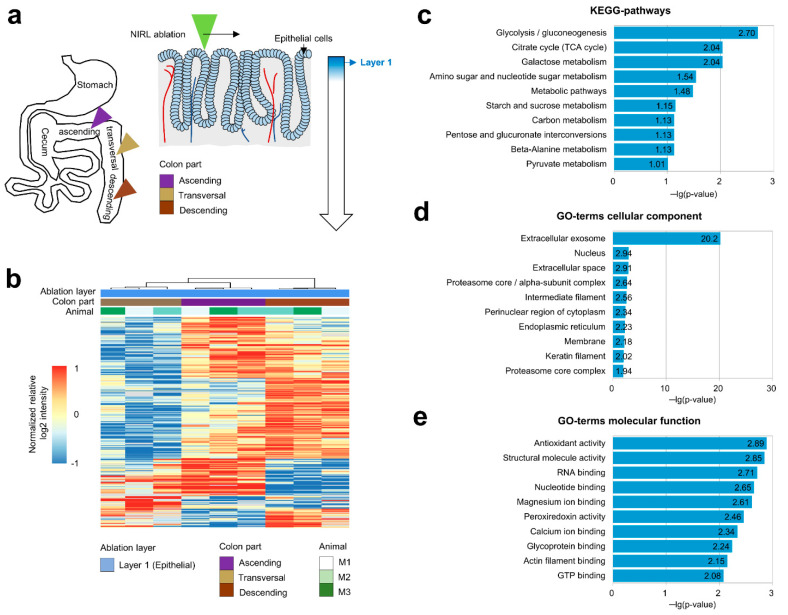
Comparison of epithelial protein signatures across different consecutive colon segments. (**a**) Schematic overview of the colon structure and the spatial location of ablated epithelial samples. (**b**) Hierarchical clustering of proteins between ascending, transversal, and descending colon epithelium. (**c**) Top 10 enriched terms (highest *p*-value) from KEGG-pathway-based DAVID enrichment of ANOVA significant proteins between ascending, transversal, and descending colon epithelium. (**d**,**e**) Top 10 enriched terms (highest *p*-value) from gene ontology (GO) cellular component-based and molecular function-based DAVID enrichment of ANOVA-significant proteins between ascending, transversal, and descending colon epithelium.

**Figure 2 ijms-23-06132-f002:**
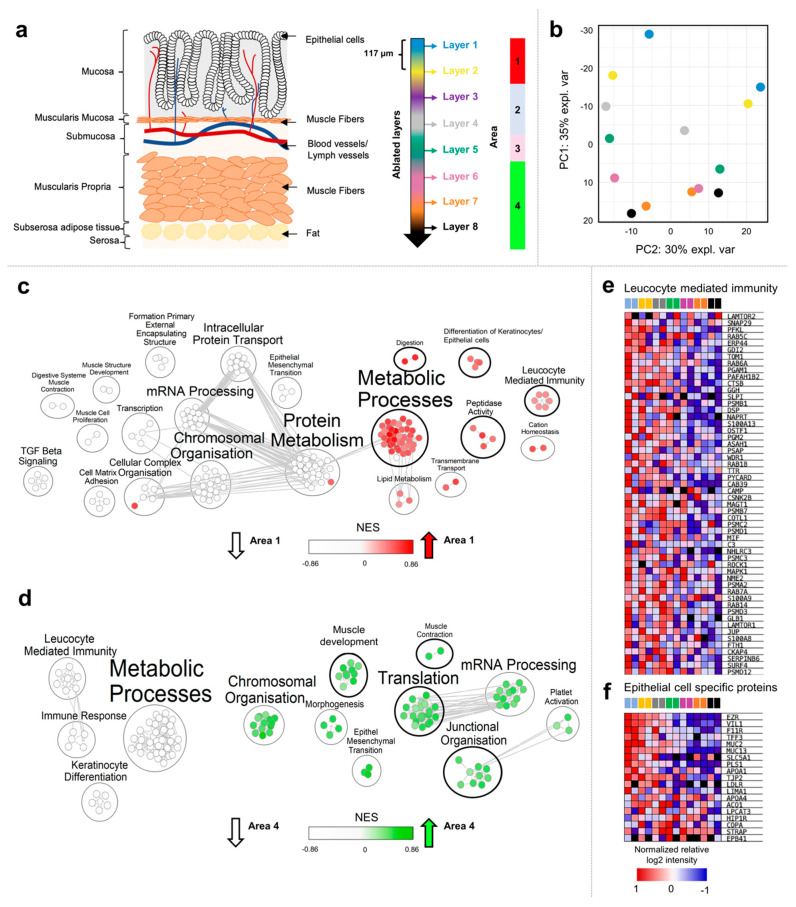
Comparison of different cellular layers of the murine intestine after NIRL ablation. (**a**) Schematic overview of the colon layer structure and the location of ablated layers and disclosed proteomic signatures along the sagittal axis of the colon. (**b**) Scatter plot visualization of the first 2 principal components from nonlinear iterative squares (NIPALS) PCA. (**c**) Positively (red) and negatively (white) enriched gene ontology biological processes (GO--PB) in area 1. (**d**) Positively (green) and negatively (white) enriched gene ontology biological processes (GO--PB) in area 4. Heatmap visualization of the relative abundance distribution of proteins assigned to the gene set “Leucocyte mediated immunity” (**e**) and to “Epithelial cell specific proteins” (**f**) across ablation layers.

**Figure 3 ijms-23-06132-f003:**
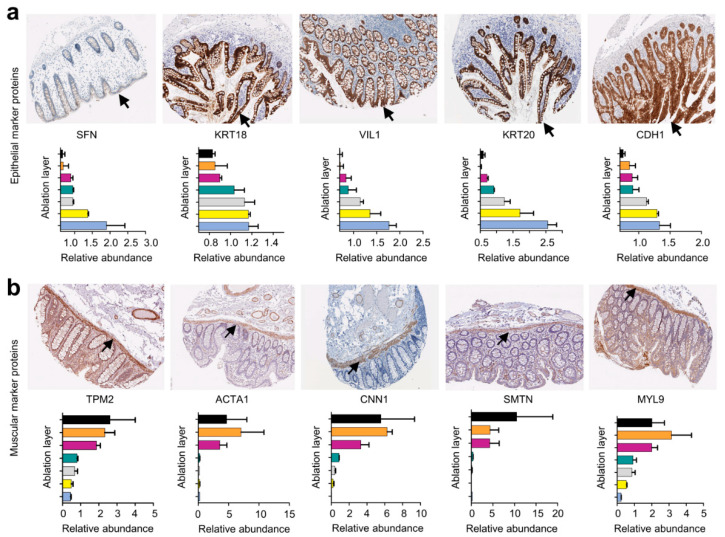
Abundance distribution of known epithelial and muscular proteins across different ablation layers. (**a**) Abundance distribution of established epithelial marker proteins (*SFN, KRT18, VIL1, KRT20*, and *CDH1*) across different ablation layers, associated with corresponding immunohistological staining of the human intestine, obtained from the Human Protein Atlas (v21.0.proteinatlas.org) [27]. (**b**) Abundance distribution of established muscular marker proteins (*TPM2, ACTA1, CNN1, SMTN,* and *MYL9*) across different ablation layers, with corresponding immunohistological staining of the human intestine, obtained from the Human Protein Atlas (v21.0.proteinatlas.org) [27].

**Figure 4 ijms-23-06132-f004:**
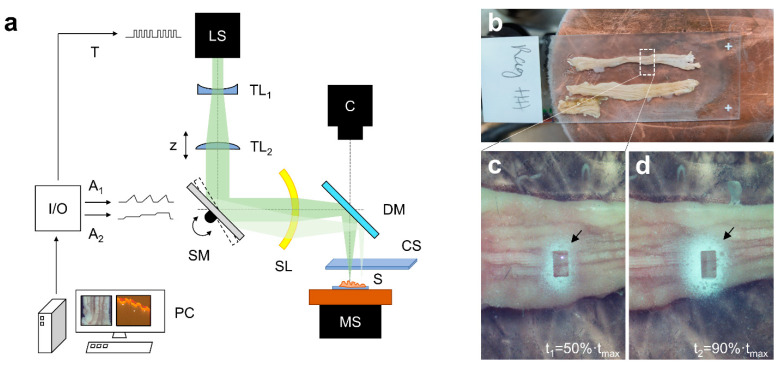
(**a**) Experimental ablation setup utilizing a nanosecond infrared laser system (LS) with a 2-axis scanning mirror (SM) in combination with a scanning lens (SL). A camera with objective (C) is monitoring the ablation of the sample (S) through a dichroitic mirror (DM). The sample is cooled down to about −10 °C and can be displaced by a manual stage (MS). The aerosol is collected by a glass slide (CS), which is placed directly above the sample. PC, control computer; I/O, input/output card; A_1/2_, analog line 1/2; T, trigger line; TL_1/2_, telescope lens 1/2. (**b**) Murine colon that is transversally cut open and placed on an objective slide for better handling. In the region of interest, the colon is sampled layer-by-layer with NIRL. (**c**,**d**) Pictures of the ablation area at two time points (t_1_ and t_2_) of the ablation time (t_max_). The trapped and dried aerosol (black arrow) appears white on the glass slide. A rectangular area within the dried aerosol is lost due to the scanning process.

**Figure 5 ijms-23-06132-f005:**
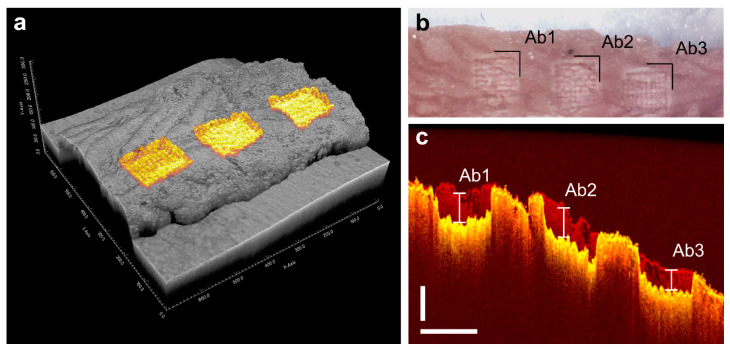
(**a**) Three-dimensional reconstruction of the two registered OCT volumes before and after the laser ablation, with the ablation surfaces highlighted in yellow. (**b**) Picture of the murine colon after the ablation of 2 layers at three positions along a line. (**c**) Example OCT brightness scan (B-scan) of the two registered volumes before (red) and after (yellow) ablation with depth measuring markings for the three ablations Ab1, Ab2, and Ab3. Scale bars measure 250 µm vertical and 500 µm horizontal, respectively.

## Data Availability

Mass spectrometric data generated in this study can be accessed through the ProteomeXchange Consortium via the PRIDE partner repository with the dataset identifier PXD033584.

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
