# Peer review of "Tissue Sampling and Homogenization with NIRL Enables Spatially Resolved Cell Layer Specific Proteomic Analysis of the Murine Intestine"

_ijms, 2022, doi:10.3390/ijms23116132_

Round 1
Reviewer 1 Report
In the current manuscript “Tissue sampling and homogenization with NIRL enables spatially resolved cell layer specific proteomic analysis of the murine intestine” by Vob et al. reported the use of nanosecond infrared laser (NIRL) coupled with LC-MS/MS proteome approach for different layers of frozen murine intestinal tissue.
Comments
- It would be interesting to list the significantly enriched proteins in each layer to confirm the identity of that layer for example layer 1 expresses most of the proteins related to epithelial cell function and layer 8 muscle/stromal cell functions.
- Figure 2E and F, heatmaps can be also generated for muscle/stromal specific proteins, intestinal stem cell specific proteins etc. to confirm technology.
Reviewer 2 Report
Voβ and coworkers reveal proteomic analysis of the murine colon tissue by using NIRL. Previously, they revealed NIRL for mass spectrometric proteomics using colon and spleen tissues in IJMS (2021, 22, 10833). Thus, the manuscript is a ver2 for NIRL for application to colon tissue in more detail. NIRL technique shown by the authors is very interesting for biological researchers as well as MS researchers. There are some comments as follows.
1) The NIRL technique and apparatus are universal or specific? Are MS researchers able to use the technique or launch the equipment?
2) Single-cell RNA sequencing (scRNA-seq) has been improved considerably during recent years and has been widely applied in stem cell biology, tumor biology, developmental biology, and so on. Using scRNA-seq, resolution provides insights into cell-specific changes in gene expression, such as cell type or state identification, trajectory inference, and the identification of therapeutic targets and biomarkers. Is it possible to combine the two technique for analyzing biomolecules (omes)?
3) Moor and coworkers performed transcriptomic analysis of laser-capture microdissected villus segments to identity landmark genes and localize single-sequenced enterocytes along the villus axis (Moor et al. Cell, 2018, 175, 116-1167). Have you ever been to analyze small intestinal tissue with NIRL? Please tell me the difference, advantage, and disadvantage between microdissction and NIRL.
4) Page9,line4: ...and control. [Joh88] -> and control [Joh88].
Author Response
Reviewer 2:
Voβ and coworkers reveal proteomic analysis of the murine colon tissue by using NIRL. Previously, they revealed NIRL for mass spectrometric proteomics using colon and spleen tissues in IJMS (2021, 22, 10833). Thus, the manuscript is a ver2 for NIRL for application to colon tissue in more detail. NIRL technique shown by the authors is very interesting for biological researchers as well as MS researchers. There are some comments as follows.
1) The NIRL technique and apparatus are universal or specific? Are MS researchers able to use the technique or launch the equipment?
Thanks to the reviewer for the comments and interest in the NIRL technique. The NIRL is a commercially available laser. The optical design of the experimental setup is specific. Trained MS researchers can use the setup. Furthermore, we are constantly improving the system regarding its capabilities, automation level and usability.
2) Single-cell RNA sequencing (scRNA-seq) has been improved considerably during recent years and has been widely applied in stem cell biology, tumor biology, developmental biology, and so on. Using scRNA-seq, resolution provides insights into cell-specific changes in gene expression, such as cell type or state identification, trajectory inference, and the identification of therapeutic targets and biomarkers. Is it possible to combine the two techniques for analyzing biomolecules (omes)?
We thank the reviewer for addressing the usability of NIRL-based sampling for the spatially resolved analysis of other biomolecule classes. Infrared laser-based sampling is highly compatible with the analysis of RNA from tissue. This issue was addressed by Wang et al. in 20191. Especially for RNA, it is challenging to maintain the analytes integrity in the tissue homogenization and cell lysis process. This is mainly caused by the rapid degradation of RNA by RNAses, Hence, a fast and RNAse free sample processing is required. This cannot be guaranteed by most methods for tissue sampling, including laser capture microdissection, which is the current state of the art for the spatially resolved sampling of RNA. Wang et al. showed that Nanosecond infrared laser (NIRL) ablation sufficiently extracted high quality intact RNA from tissue, that could be successfully converted to cDNA and analyzed by subsequent qPCR. Another advantage of infrared laser ablation, besides the elimination of time-consuming tissue homogenization procedures and spatial resolution is the ability to extract RNA and proteins at the same time. The analysis of both tissue types from one sample enables a highly reliable correlation between transcriptomic and proteomic data.
While NIRL sampling could in general be highly beneficial for the analysis of RNA, as stated thought our manuscript:
“As a limitation to our current approach, while different cellular areas of the intestine could be differentiated, a single cell type resolution was not yet achieved in this study…. “
Hence, cell layer specific analyte profiles can be obtained, while an analysis at the single cell level is yet not possible.
3) Moor and coworkers performed transcriptomic analysis of laser-capture microdissected villus segments to identity landmark genes and localize single-sequenced enterocytes along the villus axis (Moor et al. Cell, 2018, 175, 116-1167). Have you ever been to analyze small intestinal tissue with NIRL? Please tell me the difference, advantage, and disadvantage between microdissction and NIRL.
Today the spatial aspect of tissues is best addressed with laser capture microdissection (LCM). While we agree that, as for example described by Moor et al, LCM enables the high resolution spatially resolved analysis of biomolecules, the usage of LCM should be considered carefully due to the following points:
- LCM is time consuming. Sampling time varies from 20 min to 1hour for one slide. 1
- LCM requires additional homogenization and analyte extraction, which is often challenging for the small sample amounts, obtained by LCM.
èEspecially for analytes that are rapidly digested or modified, such as proteins or RNA, the analyte integrity can be reduced while sampling and homogenization with LCM.
- LCM can lead to low quality biomolecules, especially when low sample amounts of tissue are used. This for example has been shown by Wandewoestyne et al. 2 for RNA and DNA. Here DNA, extracted by infrared laser-based LCM, was not found to be suitable for reliable PCR amplification. While UV laser-based LCM enabled the analysis of DNA and RNA from low cell numbers respectively, it was found that RNA degradation was inherent to the LCM system itself.
- LCM requires fixation. Often Formalin fixated paraffin embedded (FFPE) material is used for this purpose. The usage of FFPE tissue is beneficial, as FFPE samples can be stored for a long time under cheap conditions. However, in the process of formalin fixation or the extraction of biomolecules from FFPE tissue, omes might undergo significant artificial changes. As an example Gorez et al.,3 von Ahlfen et al.4 and Pennock et al.5 describe that FFPE sample storage and tissue processing can lead to highly degraded RNA, finally resulting in limited gene detection and sequencing artifacts. For proteomics, Formalin fixation results in a limited extraction efficiency as well as the induction of irreversible chemical modification-finally leading to higher error rates and reduced protein numbers in protein identification from LC-MS/MS data. 6
The usage of NIRL ablation for the spatially resolved sampling of tissues overcomes these limitations. We thank the reviewer for allowing us to clarify this point further and cited the work of Moor at al. in our main manuscript.
4) Page9,line4: ...and control. [Joh88] -> and control [Joh88].
We thank the reviewer for his comment and changed respective parts in the manuscript accordingly. Related changes in the main text are marked in red.
- Wang, K. et al. RNA sampling from tissue sections using infrared laser ablation. Anal. Chim. Acta 1063, 91–98 (2019).
- Vandewoestyne, M. et al. Laser capture microdissection: Should an ultraviolet or infrared laser be used? Anal. Biochem. 439, 88–98 (2013).
- Groelz, D. et al. Non-formalin fixative versus formalin-fixed tissue: A comparison of histology and RNA quality. Exp. Mol. Pathol. 94, 188–194 (2013).
- von Ahlfen, S., Missel, A., Bendrat, K. & Schlumpberger, M. Determinants of RNA quality from FFPE samples. PLoS One 2, 1–7 (2007).
- Pennock, N. D. et al. RNA-seq from archival FFPE breast cancer samples: molecular pathway fidelity and novel discovery. BMC Med. Genomics 1–18 (2019).
- Magdeldin, S. & Yamamoto, T. Toward deciphering proteomes of formalin-fixed paraffin-embedded (FFPE) tissues. Proteomics 12, 1045–1058 (2012).